# Diabetes Mellitus and Periodontitis Share Intracellular Disorders as the Main Meeting Point

**DOI:** 10.3390/cells10092411

**Published:** 2021-09-13

**Authors:** Juliana Portes, Beatriz Bullón, José Luis Quiles, Maurizio Battino, Pedro Bullón

**Affiliations:** 1Department of Periodontology, Dental School, University of Seville, C/Avicena, s/n, 41009 Seville, Spain; drajulianaportes@gmail.com (J.P.); beatrizbullon@hotmail.com (B.B.); 2Biomedical Research Center (CIBM), Department of Physiology, University Campus of Cartuja, Institute of Nutrition and Food Technology “José Mataix Verdú”, University of Granada, 18071 Granada, Spain; jlquiles@ugr.es; 3Research Group on Foods, Nutritional Biochemistry and Health, Universidad Europea del Atlántico, C/Isabel Torres, 21, 39011 Santander, Spain; 4Department of Clinical Sciences, Faculty of Medicine, Università Politecnica delle Marche. Via Tronto 10A, 60126 Torrette di Ancona, Italy; m.a.battino@staff.univpm.it; 5International Research Center for Food Nutrition and Safety, Jiangsu University, 301 Xuefu Rd, Zhenjiang 212013, China

**Keywords:** periodontal disease, diabetes, mitochondrial dysfunction, oxidative stress, AMPK, autophagy

## Abstract

Diabetes and periodontitis are two of the most prevalent diseases worldwide that negatively impact the quality of life of the individual suffering from them. They are part of the chronic inflammatory disease group or, as recently mentioned, non-communicable diseases, with inflammation being the meeting point among them. Inflammation hitherto includes vascular and tissue changes, but new technologies provide data at the intracellular level that could explain how the cells respond to the aggression more clearly. This review aims to emphasize the molecular pathophysiological mechanisms in patients with type 2 diabetes mellitus and periodontitis, which are marked by different impaired central regulators including mitochondrial dysfunction, impaired immune system and autophagy pathways, oxidative stress, and the crosstalk between adenosine monophosphate-activated protein kinase (AMPK) and the renin-angiotensin system (RAS). All of them are the shared background behind both diseases that could explain its relationship. These should be taken in consideration if we would like to improve the treatment outcomes. Currently, the main treatment strategies in diabetes try to reduce glycemia index as the most important aspect, and in periodontitis try to reduce the presence of oral bacteria. We propose to add to the therapeutic guidelines the handling of all the intracellular disorders to try to obtain better treatment success.

## 1. Introduction

Diabetes is a chronic metabolic disease characterized by hyperglycemia and is one of the leading causes of death worldwide among non-communicable diseases [1]. Periodontitis is the advanced form of periodontal disease and is one of the most prevalent diseases in the world. It is mostly caused by oral microbiota dysbiosis [2,3], but some risk factors also impact its development such as diabetes, smoking and genetic predisposition [4]. Diabetes is an important grade modifier used as an indicator of the rate of periodontitis progression [5].

Inflammation marks the link between diabetes and periodontal disease [6]. However, some authors consider that there is still scarce information based on research with representative samples and prospective longitudinal studies [7]. It is important to note that it is difficult to perform a longitudinal study and to define whether the pathological associations are causal in nature in chronic inflammatory diseases studies. 

Hitherto inflammation is described as vascular and tissue alterations. An aggression produces some cytokines that induces the extravasation of plasma and blood cells that try to control and restore the damage. Periodontitis and T2DM, independently, have elevated inflammatory markers. However, when present at the same time, there is an exacerbation of this immunoinflammatory response. M1-type macrophage [8,9,10], neutrophils [11,12,13,14], and polymorphonuclear cells (PMN) [15] usually have their function upregulated, and dendritic cells are reduced or immature [16]. Consequently, pro-inflammatory cytokines such as IL-1β, IL-17, IL-6, TNFα, INFγ used to be higher and IL-10 reduced [17,18,19,20,21]. This scenario favors the great tissue destruction observed in periodontal tissues, micro and macrovascular lesions, lipid profile alterations (high low-density lipoprotein (LDL) and triglycerides) and difficulty in glycemic control [22,23,24,25].

All these aforementioned mechanisms explain the pathogenesis at the tissue level. Tissues are made up of cells that mediate the immunity and suffer the aggression. New technological advances allow us to study deeply the cell mechanisms involved. Therefore, it is essential to understand not only the pathological alterations at the tissue level, but also the intracellular molecular mechanisms involved in this process that usually occur under subclinical conditions even before a state of complete inflammatory disease is established. Preclinical studies have been helpful in understanding the basic mechanisms involved in the onset of diabetes, periodontitis, and their systemic effects [26].

Molecular inflammation may be the meeting point of such diseases responsible for perpetuating the low-grade inflammatory condition at the tissue level. This comprehensive review aims to update and emphasize the available evidence of molecular pathophysiological alterations involved in periodontitis and T2DM to help further investigate and generate new perspectives for a better clinical management of these patients, and to include them in the therapeutic and prevention guidelines to improve the outcomes.

## 2. Biological Membrane Alteration

Plasma membrane is a highly dynamic structure composed of phospholipid bilayer and lipid rafts, being one of the main structures of all living systems, delimiting cells and organelles such as lysosomes and mitochondria. Lipid rafts are composed of cholesterol, glycosphingolipids, and specific proteins, which are associated and dissociated in the second scale [27,28,29].

These microdomains are involved in cellular signaling and membrane permeability, such as endo- and exocytosis during bacterial or toxin aggression [30,31], immune cell activation [32], redox signaling [33,34], osteoclastogenesis induction [28,35], and insulin secretion and sensitivity [32,36]. The type and amount of lipids vary in each cell membrane according to their function and the individual’s diet and are influenced by lipid metabolism in health and disease condition. Disruption of this structure may alter several physiological cellular functions [27,28,29]. 

Fatty acids (FA) are important membrane structural components and signaling molecules, and any change in their length or degree of saturation can directly impact membrane plasticity. High concentration of saturated FA (SFA) induces negative effects on the plasma membrane by increasing its fluidity and activity, increasing toll-like receptor (TLR) signaling translocation and RANKL activation [28,37,38].

Patients with diabetes have high levels of SFA and overexpression of TLR4/CD36-mediated pathway in gingival fibroblasts [30,39]. Palmitate (saturated) is enhanced in hyperglycemia, and it is even higher in the presence of *P. gingivalis* [39], which is in agreement with the increase of FA uptake by lipid rafts after periodontal lipopolysaccharides (LPS) stimulation [40], suggesting an exacerbation of inflammation in individuals with T2DM and periodontitis. 

Polyunsaturated FA (PUFA) play a role in modulating mitochondrial function, inflammatory response, improving hormone sensitivity, especially insulin, and enhancing membrane fluidity and responsiveness. PUFA may present a pro-inflammatory profile (in case omega-6 prevail) in the initial phases of inflammatory response or an anti-inflammatory profile (with omega-3 being the most represented) during the resolution of inflammation. Omega-3 seems to inhibit factor nuclear kappa B (NF-κβ) activation and TLR dimerization, which reduces SFA pro-inflammatory stimulus [29,37,39,41,42], and it has been also related to clinical and immunological benefits for patients with T2DM after daily supplementation and periodontal debridement [43].

Integrity of membrane properties have been associated with a diet rich in unsaturated fats such as olive oil [44], while high-fat diet (rich in saturated fat and cholesterol) seems to alter cellular properties and exacerbate the inflammatory response and increase hyperlipidemia and alveolar bone loss in periodontitis models [45]. The inhibition of specific glycosphingolipids of lipid rafts improved glucose control, insulin sensitivity in T2DM patients [36], and prevent RANKL-osteoclast induction [33,35].

Lipid peroxidation (LPO) is an oxidative degradation of membrane lipids, which is increased in diabetes owing to the alteration of oxidative metabolism and the overproduction of reactive oxygen species (ROS) [46]. This reaction produces lipid peroxides that bind to proteins and create unstable lipid radicals. Repeated cycles of LPO can activate the NF-κβ pathway, inducing a pro-inflammatory response, contributing to maintaining oxidative stress and causing serious damage to cell membranes [46].

The intensity of LPO strongly depends on the degree of lipid unsaturation and this reaction is amplified as long as oxygen and unoxidized PUFA are available [47]. Lipid marker alterations have been associated with the severity of periodontitis and uncontrolled T2DM. Several LPO markers are used to monitor ROS production, and they are positively associated with cytokines’ local and systemic expression in patients with T2DM and periodontitis with dyslipidemia, which is even worse in poorly controlled T2DM [46] (Figure 1).

Currently, T2DM patients have dyslipidemia, an imbalance of body lipids characterized by high levels of triglycerides and LDL and low levels of high-density lipoprotein (HDL). This condition increases oxidative metabolism and LPO, thereby maintaining a vicious cycle of chronic pro-inflammatory condition [46]. The disturbance of glycemic metabolism and the continued activation of the polyol pathway to metabolize the excess of glucose also causes membrane alterations, increases LPO, and coupled with reduced antioxidant system (AOX) [48] upregulates immune cell responses and visceral adiposity [49]. Transcriptome analysis has facilitated the evidence of deregulation of different inflammatory molecular pathways, by co-expressed genes, in association with the quality of adipose tissue and type 2 diabetes [50].

## 3. Aggression Recognition

Humans are multicellular organisms, and it is essential to distinguish between our own cells and others that can be harmful, as well as physical and chemical factors, through membrane cell receptors that stimulate immunity. These are fundamental sensory elements for host defense that can be stimulated by hormones and inflammatory mediators.

The immune system recognizes aggression through the connection of pathogen-associated molecular patterns (*PAMPs*) and damage-associated molecular patterns (*DAMPs*) to pattern recognition receptors (PRRs). These PRRs can be TLRs, NOD-like receptors (NLRs), RAGE, C-type lectin receptors (CLRs), and complement receptors. The increased levels of interleukin (IL)-1β enhance the expression of some cell receptors such as TLR4 which are involved in the signaling and activation of NF-κβ and mitogen-activated protein kinase (MAPK) pathways [51].

Inflammasomes, the key regulators of innate, adaptive, and host responses, are a cytoplasmic multi-protein complex composed of NLRs and different types of proteins. NIMA-related kinase 7 (Nek7) is an indispensable upstream factor involved in NLR family pyrin domain-containing protein 3 (NLRP3) inflammasome formation and regulates the release of pro-inflammatory cytokines. The inflammasome complex activates caspase-1 and -5, which consequently release the first cytokines IL-1β and IL-18 against *PAMP* or *DAMP*, producing a cascade of local and systemic responses [52]. 

Inflammasomes are activated and modulated by different metabolic alterations, and it has been reported that *P. gingivalis* infection induces an overexpression of PRRs and NLRP3 in T2DM-periodontitis patients, along with caspase-1 and IL-1β [51,53]. The most recently discovered innate immune cells on the periodontal tissue of periodontitis patients and mice-models of periodontitis are the innate lymphoid cells (ILCs). They are activated by *PAMPs* and *DAMPs* and play a role on initiation, modulation, and resolution of inflammation through cytokine release [54]. Adenosine monophosphate-activated protein kinase (AMPK) acts as a modulator of ILCs function [54] and NLRP3 [55], reducing their negative effects.

Hyperglycemia, even in intermittent periods, exacerbates TLR4 and RAGE expression [56]. The disease severity has been related to elevations in pro-inflammatory cytokine expression and their involvement in the increased expression of RAGE or TLR4 on the surface of epithelial cells, fibroblasts, and macrophages [56]. A significant association between RAGE polymorphism and patients with periodontitis and T2DM exists, but no association was observed in patients with only periodontitis [57]. However, it is difficult to establish whether this polymorphism can be considered a risk factor related to the development of periodontitis when associated with T2DM or if this genetic alteration is just linked to diabetes. Further investigations in patients with diabetes but without periodontitis are necessary to confirm this risk [26].

Recently, polymorphisms of TNF-α, TNFR1, TNFR2 and lymphotoxin-α were evaluated: no SNP was found to be a cross-susceptibility factor between periodontitis and T2DM. Therefore, the development of periodontitis in T2DM may be related to pathological alterations in the periodontium caused by diabetes due to hyperglycemia, high AGE levels and oxidative stress. T2DM is suggested to mask the impact of periodontitis on systemic inflammation [58].

The entire transcriptional profile of LPS of *P. gingivalis* in PDL cells has been recently described, and 36 differentially expressed genes (DEGs) have been identified in PDLs cultured with LPS for 24 h and 72 h. It was possible to observe that different biological processes, molecular functions, and cellular components are involved in the initiation and progression of periodontitis [59]. Additionally, dysregulation of immunoactivation mechanisms of neutrophils and B cells were evidenced by differentially expressed genes [60,61].

## 4. Mitochondrial Dysfunction

Oxidative phosphorylation by mitochondria is responsible for most of the ATP produced and ROS production also appears [62]. ROS release at early stages is adaptative, acting as important signaling molecules after an aggression, and is controlled by intracellular redox status through AOX [52]. However, at high concentrations they cause cellular lesions [50]. 

The excess of electron donors in the mitochondrial electron transport chain is one of the main factors responsible for NADH/NAD+ redox imbalance, because as more electrons are transported, the higher the ROS production [63]. In metabolic disorders, positive feedback is established with the increased release of ROS which stimulates the neighboring mitochondria to control the excess of these molecules, resulting in more ROS production [25,62].

Mitochondrial dysfunction is considered the major source of ROS causing damage to all cellular components and disrupting the normal signaling mechanisms. Altogether, these effects directly impaired the inflammatory response, inducing a pro-inflammatory state [25,62]. Oxidative stress arises and is maintained due to the increase in mitochondrial ROS production and inefficient (or absence) of enough AOX levels, resulting in an imbalance of the cellular redox state [64,65].

Advanced glycation end products (AGEs) arise from non-enzymatic glycation and oxidation of proteins and lipids [66]. They cause cellular damage by modifying protein function and cellular interaction with the extracellular membrane, alter the intracellular Ca^2+^ concentration and mitochondrial function, deregulate the inflammatory response, influence wound repair, and increase oxidative stress through the connection with its receptors, RAGE [6,66]. It has been suggested that high levels of AGEs may modify collagen structure, making the periodontal tissues less soluble with less reparative tendency, and along with other altered cellular responses, making them more susceptible to periodontal breakdown [67].

The degradation of AGEs occurs intracellularly by endocytosis and lysosomal activity, and galectin-3 have been discovered to be an essential molecule to AGEs removal [68,69]. In addition, low levels of galectin-3 have been associated with deficiency in glucose uptake, endothelial dysfunction in a diabetic mice model [70], and increased bone loss under high glucose condition and periodontal/LPS infection [69], which are negatively regulated by micro-RNA-124-3p [69]. Patients with diabetes used to have high levels of AGE and RAGE in human gingival fibroblast which may explain the accelerated periodontitis observed in these patients in accordance with the previous studies [71,72].

Oxidative stress and the AGE-RAGE connection stimulate signaling pathways, such as MAPK and NF-κβ, with subsequent pro-inflammatory gene transcription and increased ROS production in endothelial cells, vascular smooth muscle cells, and macrophages. The high number and activity of immune cells, mostly by the excessive response of phagocytes during hyperinflammatory response, contribute to the overall cellular stress [11,65,73]. Patients with T2DM and periodontitis showed higher levels of AGEs and ROS production than healthy individuals [11]. This oxidative stress is induced even when both diseases are not present simultaneously; however, when they are together, it becomes more severe [24].

Mitochondrial dysfunction and high mitochondrial ROS production [25,62] result in cellular stress at the molecular level, causing a reduction in protein expression, loss of mitochondrial mass, and impaired membrane potential [62], and these alterations are present in diabetes and periodontitis. Moreover, the accumulation of mitochondrial DNA (mtDNA) alterations such as mtDNA heteroplasmy and copy number, noncoding ribonucleic acid (RNA), epigenetic modification of the mitochondrial genome, epitranscriptomic regulation of the mtDNA-encoded mitochondrial transcriptome and mtDNA mutations and polymorphisms have been related to endothelial dysfunction, change in metabolism of the liver, adipose tissue, myocardium, and skeletal muscles, and poor metabolic control [74,75,76,77,78]. These parameters could be used as markers to characterize the dysregulated immune-inflammatory response commonly detected in individuals with periodontitis and T2DM [25,76].

## 5. AMPK as the Central Energy Regulator

Cells use oxidative processes through catabolic reactions for energy production and defense mechanisms against bacteria and external molecules. The adenosine monophosphate-activated protein kinase (AMPK) pathway is the central regulator of intracellular energy status, and its isoform composition depends on tissue-specific genetic expression, which may explain the multiple effects of AMPK [79,80].

The AMPK has multifaceted regulatory mechanisms with largely nonoverlapping single nucleotide polymorphism (SNP) sets and it can be activated by high concentrations of AMP that blocks ATP consumption and activates ATP-generating catabolic pathways [79]. The AMPK system is also activated by high levels of Ca^2+^, ROS, hormones, drugs (metformin), and dietary polyphenols (resveratrol and anthocyanins) [81]. Emphasizing the importance of lipid metabolism beyond vascular disease and metabolic syndrome, an association between AMPK and different anthropometric and metabolic parameters has been reported, with the greatest association with adiposity and in decreasing order with the other traits: insulin secretion and resistance, plasma glucose, total/LDL cholesterol, HDL cholesterol and triglycerides [79].

The AMPK pathway has many downstream targets, such as the mammalian target of rapamycin (mTOR), ULK1 and Ang-(1–7), and it mediates significant alterations in cell metabolism and growth, such as FA and cholesterol metabolism, glucose uptake and mitochondrial biogenesis [80,82,83] and immune modulation [54]. Generally, AMPK and mTOR have counter actions [55,84,85]. Environmental disturbances, such as dysbiosis or metabolic changes in periodontitis and T2DM condition downregulates AMPK leading to an impaired immunoinflammatory response [79,83,86,87]. In addition, high levels of HDL modulate glucose metabolism and calcium-sensitive signaling cascades and activate AMPK, leading to the inhibition of adipose tissue lipolysis, reduced circulating free FA, and increased insulin secretion in T2DM patients [88,89] (Figure 2).

Countless cells are involved in wound healing, and AMPK downstream activation has been related to a reduction in the cell proliferation capacity of [55,90] and a reduction of osteogenic differentiation of human periodontal ligament (PDL) stem cells [85]. This proliferative reduction was related to delayed wound healing and tissue remodeling in patients with diabetes [90], which could explain the intense tissue destruction observed in periodontitis patients with a lower active form of AMPK [73].

Metformin has a positive effect on the AMPK/electron transport chain through the SNP target, being an important therapeutic agent in patients with poorly controlled T2DM. However, a preclinical study showed that metformin causes lysosomal perturbations in order to activate AMPK [91]. The precise mechanism behind AMPK activation by metformin is still unclear. Several studies have reported anti-inflammatory and protective effects of this substance that ameliorate not only glucose levels but also periodontal tissue destruction [55,90]. Resveratrol is another potent AMPK activator with anti-inflammatory properties. Moreover, it is a ROS scavenger that reduces nitric oxide expression and activates the AOX pathway. Improvement in bone resorption was observed after periodontal treatment plus resveratrol supplementation [92].

The renin-angiotensin system (RAS) is related to diabetes, metabolic disorders and periodontitis [80,93]. Its protective pathway (ACE2/Ang angiotensin 1–7/Mas receptor) participates in the modulation of deleterious effects of the classical pathway (ACE/Ang II/AT1 receptor) [80,94]. Recently, Ang II was related to periodontitis exacerbation by increasing TLR4 response on dendritic cells and reducing the bone loss by blocking the AT1 receptor [93]. The oral administration of Ang (1–7) or resveratrol has been shown to improve insulin sensitivity, increase lipolysis and reduce total cholesterol, triglyceride, fasting plasma glucose and resistin levels through cross-modulation between RAS and sirtuins, which increases the expression of ACE2 and AMPK [94]. Therefore, such evidence allows us to assume that there is a crosstalk between the RAS system and AMPK pathways, where the former acts as a modulator of the latter.

## 6. Cellular Debris Elimination

Cellular death or recycling pathways are activated by PRRs and/or membrane lipid signals during metabolic alterations. Autophagy [95] is responsible for the removal and recycling of unnecessary or dysfunctional cellular components and regulating energy and metabolic homeostasis through the autophagy lysosomal pathway (ALP) to keep the cells alive. This process can also inhibit or reduce inflammasome activation by removing the inflammatory mediators [96]. LC3 is a membrane marker for autophagosomes and autolysosomes [96].

Autophagy has different stages that are regulated directly by AMPK, where in its early stages it acts by activating Beclin-1 [85] and in parallel inactivating mTOR and phosphorylating ULK1, a key autophagic initiator [82,85,97], and in the final stages stimulating the conversion from LC3-I to LC3-II [85]. The early stages of autophagy have been related to an enhancement of osteogenic differentiation of PDL stem cells, however in late stages this mechanism seems to be reduced, which may indicate a suppression of autophagic activity [85]. Autophagy plays a role inducing the differentiation of PDL stem cells, which is affected by hyperglycemic environment [98].

The AMPK/mTOR pathway is also influenced by two independent systems comprised of galectins and ubiquitin. Basically, during lysosomal damage, galectin-8 inhibits mTOR, whereas galectin-9 activates AMPK [91,99], acting as important signaling molecules of membrane damage and guiding selective autophagy [91,99]. As mentioned before, metformin is widely used in diabetic treatment, and it seems to stimulate galectin-8 and -9 which may indicate that the AMPK activation mechanism is associated with lysosomal perturbations [99].

Mitophagy is a mitochondria-specific type of autophagy. The AMPK/mTOR pathway plays an important role in the selective autophagy, allowing replacement of old mitochondria in a subcellular renewal process. High glucose and palmitate levels lead to the suppression of mitophagy and turnover of autophagosomes, which may be responsible for mitochondrial dysfunction, high ROS production, and cellular senescence [82].

Alterations in autophagy mechanisms may contribute to inflammation-associated metabolic diseases and have been related to insulin resistance development and ROS-mediated autophagy by *P.gingivalis* induction [100]. *P.gingivalis* stimulation had no effect on the ALP pathway while disruption of ALP via the *ATP6VOC* gene and lysosomal acidity was observed in experimental studies after high glucose induction, which results in the accumulation of mature IL-1β, keeping the inflammatory response and increasing periodontal destruction [96].

Disruption of ALP can also lead to cell death by nonspecific degradation of cellular components or through the activation of apoptosis, the programmed cell death that stimulates proteases to degrade all the necessary cellular components. Apoptosis has an extrinsic pathway that activates caspase-8 through death receptors such as FasL and TNF-α receptors (TNFR), while the intrinsic pathway with a loss of mitochondrial membrane integrity and subsequent release of cytochrome *c* [15,101] activates caspase-9 to release Bcl-2 family members (Bax and Bak) and stimulate the JNK signaling pathway [15,101]. Caspases-3 and -8 levels were reduced in T2DM-periodontitis patients, while caspase-9 showed the same trend as that in healthy subjects [15].

Alterations in apoptosis and autophagy regulation are also related to excessive ROS production, which is considered a bridge between autophagy and apoptosis. Autophagy may have protective effects against apoptosis and excessive ROS production because these mechanisms are enhanced while autophagy is inhibited [102]. Apoptosis strongly contributes to maintaining the high levels of inflammation, bone loss induction and difficulty in periodontal tissue repair in rat models of periodontitis and T2DM [103].

PDL fibroblasts undergo apoptosis, which causes a decrease of almost 40% in its density in rats with DM and periodontitis, when compared to normoglycemic rats after periodontitis induction [103]. Furthermore, periodontal ligament cells also suffer apoptosis mediated by a complex interaction between the JNK signaling pathway and mitochondria under high levels of TNF-α, AGE, and endogenous ROS [101,102]. In contrast, a burst in bone formation with increased osteoblast cells and amounts of osteoid in non-DM subjects was observed after removal of periodontitis ligature [103,104].

The negative influence of diabetes on bone homeostasis is mediated by inflammation involving TNF-α pathways, resulting in delayed apoptosis of PMN, which in normal conditions causes spontaneous apoptosis, leading to resolution and restoration of tissue homeostasis [15,104]. The TNFR blockage on DM-periodontitis mice models reduced osteoclasts cells and apoptosis rate and increased the osteoid and new bone formation to an equivalent level of the normoglycemic group, which underpins the abovementioned results [17,104].

The effect of diabetes and periodontitis on PMN may be additive but not synergistic, and periodontal disease affects PMN apoptosis locally and systemically, which could intensify any other inflammatory condition. Interestingly, PMN-delayed apoptosis has been also related to a high body mass index and high RAGE activation [15]. In addition, apoptosis could negatively affect the dendritic cell maturation, which may justify its low levels in DM-periodontitis patients [16].

These mechanisms of cellular adaptation have been related not to the induction of cell death, but rather to an attempt to extinguish the pro-inflammatory properties of the affected cells. However, failure of these mechanisms can lead to an exacerbated inflammatory state and to the induction of necrosis. Necroptosis is involved in prolonged mitochondrial dysfunction and ROS production, along with the release of cytoplasmic content into the extracellular space, resulting in intense tissue inflammation. This form of regulated necrosis is mediated by the enzymatic activity of receptor-interacting serine/threonine-protein kinase (RIP) 1, RIP3, and mixed lineage kinase domain such as pseudokinase (MLKL-p), which can be activated by high levels of ROS, AGE, hypoxia, and TNFR signaling [105].

Diabetes-associated periodontitis patients showed necroptosis with high levels of RIP1, RIP3, and p-MLKL in addition to high levels of ROS and AGE production, and decreased expression of activating transcription factor 4 (ATF4). It was exhibited that AOX substances can upregulate ATF4 expression by scavenging ROS in hyperglycemic conditions and strongly suppressing RIP1 and RIP3 proteins, preventing necroptosis in diabetes-associated periodontitis [105].

Pyroptosis is another form of programmed cell death that occurs through the activation of caspase-1 and NLRP3, plasma membrane rupture, and the release of many pro-inflammatory cytokines. Some authors have suggested a close relationship between Nek7 and NLRP3 pathway activation and pyroptotic cell death in the onset and development of diabetes-associated periodontitis, and that metformin ameliorates the pyroptosis outcome [52] (Figure 3).

Recent evidence has shown that mitochondrial hormesis plays a role in the restoration of mitochondrial function and superoxide production via activation of the AMPK pathway. This mechanism has been related to the improvement of molecular markers of diabetes complications [30,106,107].

## 7. Conclusions

The main alterations of diabetes and periodontitis involve complex cellular aspects that explain the pathological process and justify the relationship between both conditions. Inflammation as a cellular mechanism may respond to aggression depending on your baseline statement. Plasma membrane modification, mitochondrial dysfunction, poor energy regulation, recognition of aggression, and inadequate elimination of debris may be the modified intracellular mechanisms that may explain the physiological process behind the relationship between diabetes and periodontal disease (Figure 4). All of these alter cellular metabolic homeostasis, which are essential in the shared pathophysiological process. We highlighted all of them to propose their inclusion in future studies of therapeutic and prevention guidelines of both diseases in an attempt to improve the outcomes.

## Figures and Tables

**Figure 1 cells-10-02411-f001:**
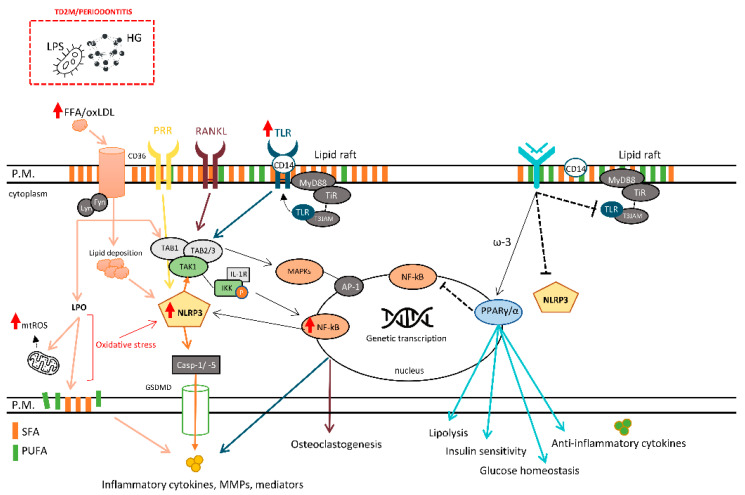
The role of plasma membrane and its influence on different cellular responses. LPS, lipopolysaccharides; FFA, free fatty acid; oxLDL, oxidized LDL; LPO, lipid peroxidation; mtROS, mitochondrial reactive oxygen species; CD14/36, cluster of differentiation 14/36; PRR, pattern recognition receptors; RANKL, receptor activator of nuclear factor-kappa beta ligand; TLR, toll-like receptor; IKK, inhibitor of kappa B kinase; IL-1R, interleukin-1 receptor; MyD88, myeloid differentiation primary response 88; NF-κB, nuclear factor kappa B; TAB1/2/3, transforming growth factor beta (TGF-β) activated kinase 1-binding protein 1/2/3; TAK1, TGF-β activated kinase 1; NLRP3, NLR family pyrin domain-containing protein 3 inflammasome; Casp-1/-5, caspase-1/5; MAPK, mitogen-activated protein kinases; AP-1, Activator protein 1; MMPs, matrix metalloproteinases; ω-3, ômega-3 polyunsaturated fatty acid; PUFA, polyunsaturated fatty acid; SFA, saturated fatty acid; HG, high glucose.

**Figure 2 cells-10-02411-f002:**
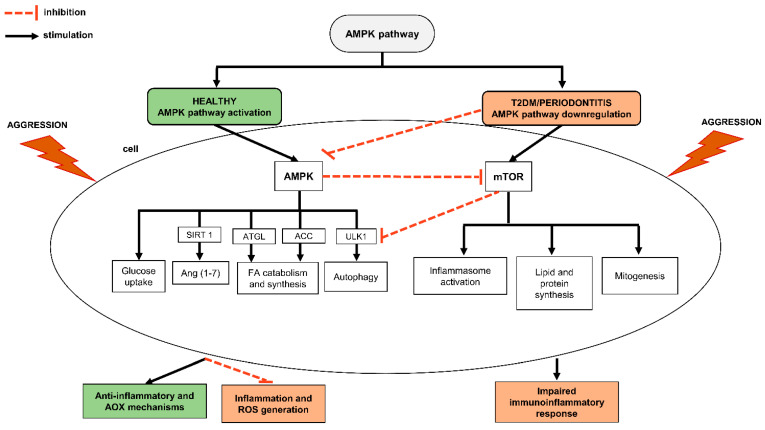
The role of AMPK pathway in healthy and diabetic periodontitis patients. AMPK has several downstream mechanisms. The balance between AMPK and mTOR in patients with T2DM and periodontitis is altered and directly influences in cellular response, increasing inflammatory response, mitogenesis and blockage of autophagy. Ang (1–7), angiotensin (1–7); AOX, antioxidants; SIRT1, sirtuins 1; GLUT4, glucose transporter type 4; ATGL, adipose triglyceride lipase; ACC, acetyl-CoA carboxylase; ULK1, unc-51 like autophagy activating kinase 1.

**Figure 3 cells-10-02411-f003:**
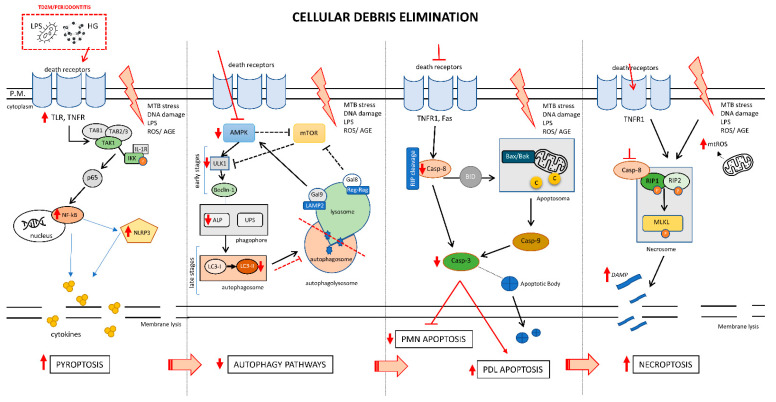
The role of cellular stress adaptation and debris eliminations. Healthy cells with normal metabolism have different resources for eliminating debris after an aggression with the aim of keeping the inflammatory response controlled. In relation to patients with T2DM/periodontitis and considering their abnormal metabolism, these mechanisms are altered leading to different responses, depending on the duration of this inflammatory state that leads to an impaired immunoinflammatory response and increased tissue destruction. MTB, metabolic; LPS, lipopolysaccharides; UBS, ubiquitin-proteasome system; LAMP2, Lysosomal Associated Membrane Protein 2; Casp-8/9/3, caspase 8/9/3.

**Figure 4 cells-10-02411-f004:**
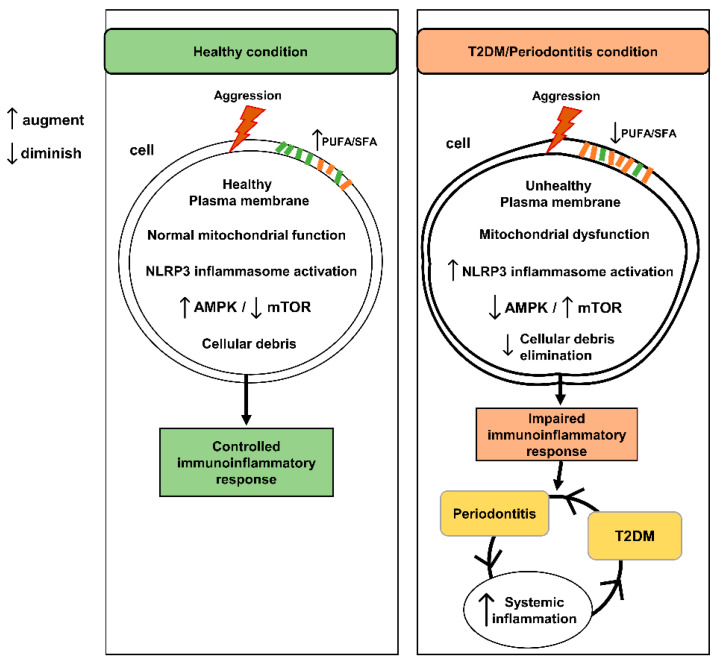
The role of the altered intracellular mechanisms behind the relationship between T2DM and periodontitis. Plasma membrane alterations, mitochondrial dysfunction, imbalance of AMPK/mTOR pathway, excessive NLRP3 activation and disruption of the debris elimination mechanisms as the critical point to maintain the chronic systemic inflammation.

## Data Availability

The study did not report any data.

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
