# Peer review of "Diabetes Mellitus and Periodontitis Share Intracellular Disorders as the Main Meeting Point"

_cells, 2021, doi:10.3390/cells10092411_

Round 1
Reviewer 1 Report
The manuscript made by Bullon P et al, describes the relationship between periodontitis and T2DM regarding molecular interactions on several pathways, focusing mainly on the interaction between inflammation by periodontitis and T2DM. Despite this review is interesting, the authors describe superficially the cellular interaction of periodontitis and T2DM. Moreover, they describe several cell interactions focusing on inflammation process, but they do not focus on the main subject (T2DM and periodontitis).
Why do not they describe first the main biological aspects of periodontitis in patients with diabetes? On the point 2: In this point (Biological membrane alteration) authors describe generally the interaction of FA, PUFA and LPO with low consistence about T2DM and periodontitis.
On the point 3: Mitochondrial dysfunction. The authors describe an important event related to immune cell function associated with AGE and RAGE, but they do not describe the possible interaction about hyperglycaemia with AGE/RAGE and PDL (periodontal ligament) and how these interactions are related with gingival tissue, fibroblasts, and interleukins. I consider the authors need to deepen about this interesting subject and, if possible, to make a diagram in which these interactions can be explained (AGE/RAGE/PDL/T2DM/periodontitis).
On the point 4: This is an interesting paragraph about interaction of AMPK and T2DM/periodontitis. I recommend to explain what is the relationship about AMPK and: autophagy associated with T2DM/periodontitis, glycogen metabolism and interaction with lysosomes.
On point 5, the authors need to deepen on the relationship with ecosystem associated to periodontitis and T2DM.
The conclusions are scarce and do not provide any hypotheses, nor explain any related proposal to "What is Left to Know?
Overall, this manuscript is interesting since the authors try to explain the cellular and molecular interaction associated with periodontitis and T2DM, but they do not deepen in the main subject (T2DM/diabetes). The manuscript does not have any diagram, scheme or table relating the main subject with several events that the authors explain.
Author Response
At first, we would like to thanks all the amendments proposed by the reviewer that allow us to improve the manuscript.
The manuscript made by Bullon P et al, describes the relationship between periodontitis and T2DM regarding molecular interactions on several pathways, focusing mainly on the interaction between inflammation by periodontitis and T2DM. Despite this review is interesting, the authors describe superficially the cellular interaction of periodontitis and T2DM. Moreover, they describe several cell interactions focusing on inflammation process, but they do not focus on the main subject (T2DM and periodontitis).
Our main point is not to describe the cellular interactions but the meeting point between both of them. Our idea is that the patients suffering both diseases have a shared pathological background: the inflammatory intracellular mechanisms. Trying to highlight this aim we change the title, the abstract and the conclusions emphasizing and highlighting this aspect.
Also, we change the cover letter:
Diabetes mellitus and periodontitis are two of the most prevalent diseases that are related based on epidemiological studies. It has been stated that both can influence the other one impairing the severity and prognostic. It seems that there are two independent pathological processes. But there are strongly related through inflammation as the meeting point. Inflammation is a defense mechanism starting with an aggression, so vascular and cellular components try to control it. But mainly inflammation involves intracellular mechanisms. Different metabolic and cellular organelles disorders impair the way of all the cells to deal with all types of aggressions. These are shared by diabetes and periodontitis. We review plasma membrane modification, mitochondrial dysfunction, poor energy regulation, recognition of aggression, and inadequate elimination of debris. Most of them are impaired in both diseases and should be the basic pathological statement that should be taken in consideration if we would like to improve the treatment outcomes. Nowadays, the main treatment strategies in diabetes try to reduce glycemia index as the most important aspect and in periodontitis try to reduce the presence of oral bacteria. We propose to add in the therapeutic guidelines the handling of all the intracellular disorders trying to obtain better success.
Why do not they describe first the main biological aspects of periodontitis in patients with diabetes?
Our goal was not to describe the main biological aspects but the main intracellular mechanisms behind those diseases.
On the point 2: In this point (Biological membrane alteration) authors describe generally the interaction of FA, PUFA and LPO with low consistence about T2DM and periodontitis.
We change the text according to the suggestion. We added more information based on preclinical studies that evidence the influence of the incorporation of FA, according to its degree of saturation, in individuals with periodontitis and diabetes.
On the point 3: Mitochondrial dysfunction. The authors describe an important event related to immune cell function associated with AGE and RAGE, but they do not describe the possible interaction about hyperglycaemia with AGE/RAGE and PDL (periodontal ligament) and how these interactions are related with gingival tissue, fibroblasts, and interleukins. I consider the authors need to deepen about this interesting subject and, if possible, to make a diagram in which these interactions can be explained (AGE/RAGE/PDL/T2DM/periodontitis).
As we mentioned previously, we do not review the specific interaction of hyperglycemia and PDL but the meeting point between T2DM and periodontitis in the intracellular metabolic level.
On the point 4: This is an interesting paragraph about interaction of AMPK and T2DM/periodontitis. I recommend to explain what is the relationship about AMPK and: autophagy associated with T2DM/periodontitis, glycogen metabolism and interaction with lysosomes.
Thank you for your point but we think these aspects are explained in the point 6
On point 5, the authors need to deepen on the relationship with ecosystem associated to periodontitis and T2DM.
All the elements of the ecosystems have a common way of recognition through the PAMPs, DAMPs, PRRs and inflammasomes. We have considered that the description of the ecosystems associated with periodontitis and T2DM could be another review and was not our aim.
The conclusions are scarce and do not provide any hypotheses, nor explain any related proposal to "What is Left to Know?
We change the conclusions and the title to highlight the importance to take in account these aspects in the future treatment and preventive strategies
Overall, this manuscript is interesting since the authors try to explain the cellular and molecular interaction associated with periodontitis and T2DM, but they do not deepen in the main subject (T2DM/diabetes). The manuscript does not have any diagram, scheme or table relating the main subject with several events that the authors explain.1
Our main goal is not to propose interaction between both diseases but the main background at the intracellular level that could improve the treatment and preventive outcomes

Reviewer 2 Report
Authors describe alteration of cellular membrane, mitochondrial function, AMPK pathway, cellular sensors, intercellular and cellular debris elimination in patients and animal models of periodontitis with diabetes. The reviewer has several concerns and recommends to reconsideration of the manuscript structure.
Major comments
- The reviewer recommends to describe causes, cellular constituents of inflammation, known mechanisms of periodontitis and type 2 diabetes in introduction for the reader.
- The authors described well known-pathophysiological alterations in type 2 diabetes and extend it to periodontits in parallel ways. There is a few information on the cell types in the lesions. Therefore, it is difficult to understand special interactions and mechanistic insights linking type 2 diabetes to periodontitis. The reviewer recommends to describe altered cell type in the lesions, presence of etiological factors, their sensors, signaling pathway in the lesions of periodontitis and type 2 diabetes in sequential way.
- The title is “molecular pathophysiological mechanisms”, however, the information mainly derived from clinical studies especially in chapter 2, 3, 4. Examples for citation. (1) Schmidt AM, et al. Advanced glycation endproducts (AGEs) induce oxidant stress in the gingiva: a potential mechanism underlying accelerated periodontal disease associated with diabetes. J Periodontal Res. 1996;31:508-15. (2) Qin X, et al. Increased Innate Lymphoid Cells in Periodontal Tissue of the Murine Model of Periodontitis: The Role of AMP-Activated Protein Kinase and Relevance for the Human Condition. Front Immunol. 2017;8:922. (3) Yasunaga M, et al. The Early Autophagic Pathway Contributes to Osteogenic Differentiation of Human Periodontal Ligament Stem Cells. J Heard Tissue Biol. 2019;28:63-70.
- Authors assume similar inflammatory mechanisms in periodontitis and type 2 diabetes. However, periodontitis is a chronic infectious disease, and type 2 diabetic condition augments chronic inflammatory diseases. Although the reviewer agrees with the vicious circle between them, and I recommend to describe what, where, and how any molecular factors affect the augmentation of periodontitis under type 2 diabetic condition. Please describe the molecular linkage on the pathophysiology in detail. In addition, are there any molecular differences between periodontitis and type 2 diabetes?
- How about any single cell RNA-seg and proteome data on periodontitis to understand molecular and cellular interactions?
Author Response
At first, we would like to thanks all the amendments proposed by the reviewer that allow us to improve the manuscript
Authors describe alteration of cellular membrane, mitochondrial function, AMPK pathway, cellular sensors, intercellular and cellular debris elimination in patients and animal models of periodontitis with diabetes. The reviewer has several concerns and recommends to reconsideration of the manuscript structure.
Major comments
- The reviewer recommends to describe causes, cellular constituents of inflammation, known mechanisms of periodontitis and type 2 diabetes in introduction for the reader.
We modify the text according to the suggestion
- The authors described well known-pathophysiological alterations in type 2 diabetes and extend it to periodontits in parallel ways. There is a few information on the cell types in the lesions. Therefore, it is difficult to understand special interactions and mechanistic insights linking type 2 diabetes to periodontitis. The reviewer recommends to describe altered cell type in the lesions, presence of etiological factors, their sensors, signaling pathway in the lesions of periodontitis and type 2 diabetes in sequential way.
Our main goal is not to describe the interactions between both diseases, there are previous paper describing these issues. Our proposal is to highlight all the shared intracellular mechanisms that are the main alterations in the patients that can suffer one or both diseases. We consider that is the best way to improve our treatment outcomes and prevention if we incorporate this point of view in our clinical management
- The title is “molecular pathophysiological mechanisms”, however, the information mainly derived from clinical studies especially in chapter 2, 3, 4. Examples for citation. (1) Schmidt AM, et al. Advanced glycation endproducts (AGEs) induce oxidant stress in the gingiva: a potential mechanism underlying accelerated periodontal disease associated with diabetes. J Periodontal Res. 1996;31:508-15. (2) Qin X, et al. Increased Innate Lymphoid Cells in Periodontal Tissue of the Murine Model of Periodontitis: The Role of AMP-Activated Protein Kinase and Relevance for the Human Condition. Front Immunol. 2017;8:922. (3) Yasunaga M, et al. The Early Autophagic Pathway Contributes to Osteogenic Differentiation of Human Periodontal Ligament Stem Cells. J Heard Tissue Biol. 2019;28:63-70.
We introduce the mentioned references
- Authors assume similar inflammatory mechanisms in periodontitis and type 2 diabetes. However, periodontitis is a chronic infectious disease, and type 2 diabetic condition augments chronic inflammatory diseases. Although the reviewer agrees with the vicious circle between them, and I recommend to describe what, where, and how any molecular factors affect the augmentation of periodontitis under type 2 diabetic condition. Please describe the molecular linkage on the pathophysiology in detail. In addition, are there any molecular differences between periodontitis and type 2 diabetes?
Our main point is that both diseases share inflammatory intracellular mechanisms. We agree that there are interactions between them. But there is a basic pathologic statement that alter the way they face the aggression. In periodontitis it starts due to an infection caused by bacteria and in diabetes caused by metabolic disorder and maybe by bacteria. To deal with both diseases we have to take in account these aspects. Maybe to improve our outcomes in the treatment and prevention should emphasize on it.
- How about any single cell RNA-seg and proteome data on periodontitis to understand molecular and cellular interactions?
We agree that it is a very interesting point but could be the subject of another review that cover all the published data.

Round 2
Reviewer 1 Report
The manuscript has improved considerably in comparison to the prior version. Now is more understandable and better explained.
Author Response
The manuscript has improved considerably in comparison to the prior version. Now is more understandable and better explained.
Thank you for your support
Reviewer 2 Report
- Author's proposal oversimplifies the molecular mechanisms of the diseases. Human is not a single cell. For example, in figure 1, glut 4 mainly localizes in adipocytes and striated muscle cells.
- Author's response are insufficient for the previous comments.(1)Each study shows clinical and basic experimental findings under a situation. The reviewer recommends to describe the situation. (2) The reviewer recommends to describe the evidences in sequential way for understanding of molecular mechanisms. (3) Single cell RNA seq. and transcriptome analysis are recent and strong tools for understanding of molecular mechanisms.
- There is a lack of > reference 73.
Author Response
Reviewer 2
- Author's proposal oversimplifies the molecular mechanisms of the diseases. Human is not a single cell. For example, in figure 1, glut 4 mainly localizes in adipocytes and striated muscle cells.
We have changed fig 1 according to your suggestion. We do not oversimplify the molecular mechanism we try to highlight this aspect pf pathophysiology of these diseases
- Author's response are insufficient for the previous comments.(1)Each study shows clinical and basic experimental findings under a situation. The reviewer recommends to describe the situation. (2) The reviewer recommends to describe the evidences in sequential way for understanding of molecular mechanisms. (3) Single cell RNA seq. and transcriptome analysis are recent and strong tools for understanding of molecular mechanisms.
(1) We have described the situation taken in account the limit of number of words
(2) We try to describe a sequential way and add a new figure
(3) We incorporate single cell RNA seq. and transcriptome references
- There is a lack of > reference 73.
We correct the lack of reference 73